# Global Convergence of Gradient Descent
# for Deep Linear Residual Networks

**Lei Wu**[*]   **Qingcan Wang**[*]   **Chao Ma**
Program in Applied and Computational Mathematics
Princeton University
Princeton, NJ 08544, USA
{leiwu,qingcanw,chaom}@princeton.edu

## Abstract

We analyze the global convergence of gradient descent for deep linear residual networks by proposing a new initialization: zero-asymmetric (ZAS) initialization. It is motivated by avoiding stable manifolds of saddle points. We prove that under the ZAS initialization, for an arbitrary target matrix, gradient descent converges to an $\varepsilon$-optimal point in $O\left(L^3 \log(1/\varepsilon)\right)$ iterations, which scales polynomially with the network depth $L$. Our result and the $\exp(\Omega(L))$ convergence time for the standard initialization (Xavier or near-identity) [18] together demonstrate the importance of the residual structure and the initialization in the optimization for deep linear neural networks, especially when $L$ is large.

## 1 Introduction

It is widely observed that simple gradient-based optimization algorithms are efficient for training deep neural networks [21], whose landscape is highly non-convex. To explain the efficiency, traditional optimization theories cannot be directly applied and the special structures of neural networks must be taken into consideration. Recently many researches are devoted to this topic [13, 21, 4, 7, 6, 1, 23, 15, 16], but the theoretical understanding is still far from sufficient.

In this paper, we focus on a simplified case: the deep linear neural network

$$f(\boldsymbol{x}; W_1, \ldots, W_L) = W_L W_{L-1} \cdots W_1 \boldsymbol{x}, \qquad (1.1)$$

where $W_1, \ldots, W_L$ are the weight matrices and $L$ is the depth. Linear networks are simple since they can only represent linear transformation, but they preserve one of the most important aspects of deep neural networks, the layered structure. Therefore, analysis of linear networks will be helpful for understanding nonlinear cases. For example, the random orthogonal initialization proposed in [17] that analyzes the gradient descent dynamics of deep linear networks was later shown to be useful for training recurrent networks with long term dependences [19].

Despite the simplicity, the optimization of deep linear neural networks is still far from being well understood, especially the global convergence. [18] proves that the number of iterations required for convergence could scales exponentially with the depth $L$. The result requires two conditions: (1) the width of each layer is 1; (2) the gradient descent starts from the standard Xavier [9] or near-identity [11] initialization. If these conditions break, the negative results does not imply that gradient descent cannot efficiently learn deep linear networks in general. [5] shows that if the width of every layer increases with the network depth, gradient descent with the Gaussian random initialization does find the global minima while the convergence time only scales polynomially with the depth. Here we attempt to circumvent the negative result in [18] by using better initialization strategies instead of increasing the width.

---

[*]Equal contribution

**Our Contributions**   We propose the *zero-asymmetric (ZAS) initialization*, which initializes the output layer $W_L$ to be zero and all the other layers $W_l$, $l = 1, \ldots, L-1$ to be identity. So it is a *linear residual network* with all the residual blocks and the output layer being zero. We then analyze how the initialization affects the gradient descent dynamics.

- We prove that starting from the ZAS initialization, the number of iterations required for gradient descent to find an $\varepsilon$-optimal point is $O\left(L^3 \log(1/\varepsilon)\right)$. The only requirement for the network is that the width of each layer is not less than the input dimension and the result applies to arbitrary target matrices.

- We numerically compare the gradient descent dynamics between the ZAS and the near-identity initialization for multi-dimensional deep linear networks. The comparison clearly shows that the convergence of gradient descent with the near-identity initialization involves a saddle point escape process, while the ZAS initialization never encounters any saddle point during the whole optimization process.

- We provide an extension of the ZAS initialization to the nonlinear case. Moreover, the numerical experiments justify its superiority compared to the standard initializations.

## 1.1   Related work

**Linear networks**   The first line of works analyze the whole landscape. The early work [3] proves that for two-layer linear networks, all the local minima are also global minima, and this result is extended to deep linear networks in [13, 14]. [10] provides a simpler proof of this result for deep residual networks, and shows that the Polyak-Łojasiewicz condition is satisfied in a neighborhood of a global minimum. However, these results do not imply that gradient descent can find global minima, and also cannot tell us the number of iterations required for convergence.

The second line of works directly deal with the trajectory of gradient descent dynamics, and our work lies in this venue. [17] provides an analytic analysis to the gradient descent dynamics of linear networks, which nevertheless does not show that gradient descent can find global minima. [12] studies the properties of solutions that the gradient descent converges to, without providing any convergence rate. [4, 2] consider the following simplified objective function for whitened data,

$$\mathcal{R}(W_1, \ldots, W_L) = \frac{1}{2}\|W_L \cdots W_1 - \Phi\|_F^2.$$

Specifically, [4] analyzes the convergence of gradient descent with the identity initialization: $W_L = \cdots = W_1 = I$, and proves that if the target matrix $\Phi$ is positive semi-definite or the initial loss is small enough, a polynomial-time convergence can be guaranteed. [2] extends the analysis to more general target matrices by imposing more conditions on the initialization: (1) approximately balance condition, $\|W_{l+1}^{\mathsf{T}} W_{l+1} - W_l W_l^{\mathsf{T}}\|_F \leq \delta$; (2) rank-deficient condition, $\|W_L \cdots W_1 - \Phi\|_F \leq \sigma_{\min}(\Phi) - c$ for a constant $c > 0$. The condition (2) still requires small initial loss, thus the convergence is local in nature. As a comparison, we do not impose any assumption on the target matrix or the initial loss.

As mentioned above, our work is closely related to [18], which proves that for one-dimensional deep linear networks, gradient descent with the standard Xavier or near-identity initialization requires at least $\exp(\Omega(L))$ iterations for fitting the target matrix $\Phi = -I$. However, our result shows that this difficulty can be overcome by adopting a better initialization. [5] shows that if the width of each layer is larger than $\Omega(L \log(L))$, then gradient descent converges to global minima at a rate $O(\log(1/\varepsilon))$. As a comparison, our result only requires that the width of each layer is not less than the input dimension.

**Nonlinear networks**   [6, 1, 23] establish the global convergence for deep networks with the width $m \geq \text{poly}(n, L)$, where $n$ denotes the number of training examples. [8] proves a similar result but for specific neural networks with long-distance skip connections, which only requires the depth $L \geq \text{poly}(n)$ and the width $m \geq d + 1$, where $d$ is the input dimension.

The ZAS initialization we propose also closely resembles the "fixup initialization" recently proposed in [22]. Therefore, our result partially provides a theoretical explanation to the efficiency of fixup initialization for training deep residual networks.

## 2 Preliminaries

Given training data $\{(\boldsymbol{x}_i, \boldsymbol{y}_i)\}_{i=1}^n$ where $\boldsymbol{x}_i \in \mathbb{R}^{d_{\boldsymbol{x}}}$ and $\boldsymbol{y}_i \in \mathbb{R}^{d_{\boldsymbol{y}}}$, a linear neural network with $L$ layers is defined as

$$f(\boldsymbol{x}; W_1, \ldots, W_L) = W_L W_{L-1} \cdots W_1 \boldsymbol{x}, \tag{2.1}$$

where $W_l \in \mathbb{R}^{d_l \times d_{l-1}}$, $l = 1, \ldots, L$ are parameter matrices, and $d_0 = d_{\boldsymbol{x}}$, $d_L = d_{\boldsymbol{y}}$. Then the least-squares loss

$$\tilde{\mathcal{R}}(W_1, \ldots, W_L) \stackrel{\text{def}}{=} \frac{1}{2} \|W_L W_{L-1} \cdots W_2 W_1 X - Y\|_F^2, \tag{2.2}$$

where $X = (\boldsymbol{x}_1, \boldsymbol{x}_2, \ldots, \boldsymbol{x}_n) \in \mathbb{R}^{d_{\boldsymbol{x}} \times n}$ and $Y = (\boldsymbol{y}_1, \boldsymbol{y}_2, \ldots, \boldsymbol{y}_n) \in \mathbb{R}^{d_{\boldsymbol{y}} \times n}$.

Following [4, 2], in this paper we focus on the following simplified objective function

$$\mathcal{R}(W_1, \ldots, W_L) \stackrel{\text{def}}{=} \frac{1}{2} \|W_L W_{L-1} \cdots W_2 W_1 - \Phi\|_F^2, \tag{2.3}$$

where $W_l \in \mathbb{R}^{d \times d}$, $l = 1, \ldots, L$ and $\Phi \in \mathbb{R}^{d \times d}$ is the target matrix. Here we assume $d_l = d$, $l = 1, \ldots, L$ for simplicity.

The gradient descent is given by

$$W_l(t + 1) = W_l(t) - \eta \nabla_l \mathcal{R}(t), \quad l = 1, \ldots, L, \ t = 0, 1, 2, \ldots \tag{2.4}$$

In the following, we will always use the index $t$ to denote the value of a variable after the $t$-th iteration. $\nabla_l \mathcal{R}$ is the gradient of $\mathcal{R}$ with respect to the weight matrix $W_l$:

$$\nabla_l \mathcal{R} \stackrel{\text{def}}{=} \frac{\partial \mathcal{R}}{\partial W_l} = W_{L:l+1}^\mathsf{T}(W_{L:1} - \Phi)W_{l-1:1}^\mathsf{T},$$

where $W_{l_2:l_1} \stackrel{\text{def}}{=} W_{l_2} W_{l_2-1} \cdots W_{l_1+1} W_{l_1}$. Moreover, we keep the learning rate $\eta > 0$ fixed for all iterations.

**Notations** In matrix equations, let $I$ and $0$ be the $d$-dimensional identity matrix and zero matrix respectively. Let $\lambda_{\min}(S)$ be the minimal eigenvalue of a symmetric matrix $S$ and $\sigma_{\min}(A)$ be the minimal singular value of a square matrix $A$. Let $\|A\|_F$ and $\|A\|_2$ be the Frobenius norm and $\ell_2$ norm of matrix $A$ respectively. Recall that $A(t)$ denotes the value of any variable $A$ after the $t$-th iteration, and $\nabla_l \mathcal{R}$ is the gradient of $\mathcal{R}$ with respect to the weight matrix $W_l$. We use standard notation $O(\cdot)$ and $\Omega(\cdot)$ to hide constants independent of network depth $L$.

## 3 Zero-asymmetric initialization

In this section, we first describe the zero-asymmetric initialization, and then illustrate by a simple example why this special initialization is helpful for optimization.

**Definition.** For deep linear neural network (2.3), define the *zero-asymmetric (ZAS) initialization* as

$$W_l(0) = I, \ l = 1, \ldots, L - 1, \quad \text{and} \quad W_L(0) = 0. \tag{3.1}$$

Under the ZAS initialization, the function represented by the network is a zero matrix. While commonly used initialization such as the Xavier and the near-identity initialization treats all the layers equally, our initialization takes the output layer differently. In this sense, we call the initialization asymmetric.

Let $W_l = I + U_l, l = 1, \ldots, L - 1$, then the linear network has the residual form

$$\mathcal{R} = \frac{1}{2} \|W_L(I + U_{L-1}) \cdots (I + U_1) - \Phi\|_F^2.$$

Since $\partial \mathcal{R}/\partial U_l = \partial \mathcal{R}/\partial W_l$, the dynamics will be the same as ZAS if we initialize $U_l(0) = W_L(0) = 0$. Therefore, ZAS is equivalent to initializing all the residual blocks and the output layer with zero in a linear residual network. From this perspective, the ZAS initialization closely resembles the "fixup initialization" [22] for nonlinear ResNets.

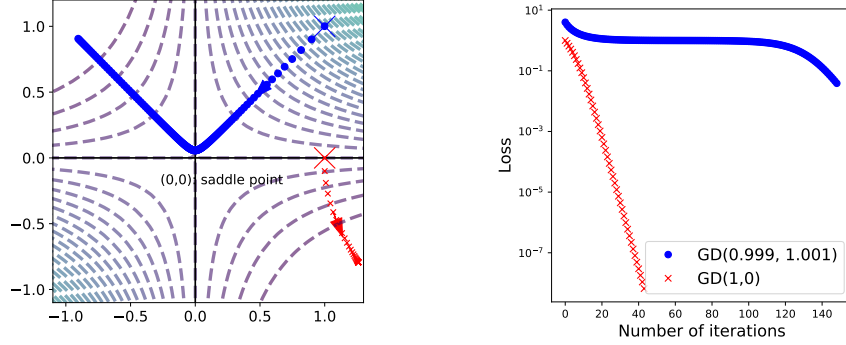

Figure 1: **(Left)** The landscape of the toy model $\mathcal{R}(w_1, w_2)$ and the two gradient descent trajectories. **(Right)** The dynamics of loss for two gradient descent trajectories. The blue curve is the gradient descent trajectory initialized from $(1-0.001, 1+0.001)$ (near-identity), and the red curve corresponds to the ZAS initialization $(1, 0)$. We observe that the blue curve takes a long time in the neighborhood of saddle point $(0, 0)$, however the red curve does not.

**Understanding the role of initialization**   Following [18], consider the following optimization problem for one-dimensional linear network with target $\Phi = -1$:

$$\mathcal{R}(w_1, w_2, \ldots, w_L) = (w_L w_{L-1} \cdots w_1 + 1)^2 / 2. \tag{3.2}$$

The origin $O(0, \ldots, 0)$ is a saddle point of $\mathcal{R}$, so gradient descent with small initialization, e.g., Xavier initialization, will spend long time escaping the neighborhood of $O$. In addition,

$$\mathcal{M} = \{(w_1, \ldots, w_L) : w_1 = w_2 = \cdots = w_L \geq 0\}$$

is a stable manifold of $O$, i.e., gradient flow starting from any point in $\mathcal{M}$ will converge to $O$. The near-identity initialization introduces perturbation to leave $\mathcal{M}$: $w_l(0) \sim \mathcal{N}(1, \sigma^2), l = 1, \ldots, L$ for some small $\sigma$. However, [18] proves that it will still be attracted to the neighborhood of $O$, thus cannot guarantee the polynomial-time converge. As a comparison, the ZAS initialization breaks the symmetry by initialize the output layer to be 0.

Figure 1 provides a numerical result for depth $L = 2$. The near-identity initialization (blue curve) spends long time escaping the saddle region, while the ZAS initialization (red curve) converges to the global minima without attraction by the saddle point.

## 4   Main results

We first provide and prove the continuous version of our main convergence result, i.e., the limit dynamics when $\eta \to 0$. Then we give the result for discrete gradient descent, whose detailed proof is left to the appendix.

### 4.1   Continuous-time gradient descent

The continuous-time gradient descent dynamics is given by

$$\dot{W}_l(t) = -\nabla_l \mathcal{R}(t), \quad l = 1, \ldots, L, \, t \geq 0. \tag{4.1}$$

In this section, we always denote $\dot{A}(t) = dA(t)/dt$ for any variable $A$ depending on $t$. For the continuous dynamics, we have the following convergence result.

**Theorem 4.1** (Continuous-time gradient descent)**.** *For the deep linear network* (2.3)*, the continuous-time gradient descent* (4.1) *with the zero-asymmetric initialization* (3.1) *satisfies*

$$\mathcal{R}(t) \leq e^{-2t}\mathcal{R}(0), \quad t \geq 0, \tag{4.2}$$

*for any $\Phi \in \mathbb{R}^{d \times d}$ and $L \geq 1$.*

The theorem above holds for arbitrary $\Phi$, and does not require depth or width to be large. To prove the theorem, we first define a group of invariant matrices as following. Note that they also play a key role in the analysis of [2].

**Definition.** For a deep linear network (2.3), define the *invariant matrix*

$$D_l = W_{l+1}^\mathsf{T} W_{l+1} - W_l W_l^\mathsf{T}, \quad l = 1, 2, \ldots, L-1. \tag{4.3}$$

**Lemma 4.2.** *The invariant matrices* (4.3) *are indeed invariances under continuous-time gradient descent* (4.1), *i.e.,* $D_l(t) = D_l(0)$ *for* $l = 1, \ldots, L-1$ *and* $t \geq 0$.

*Proof.* Recall that

$$\dot{W}_l = -\nabla_l \mathcal{R} = -W_{L:l+1}^\mathsf{T}(W_{L:1} - \Phi)W_{l-1:1}^\mathsf{T},$$

we have

$$\dot{W}_l W_l^\mathsf{T} = -W_{L:l+1}^\mathsf{T}(W_{L:1} - \Phi)W_{l:1}^\mathsf{T} = W_{l+1}^\mathsf{T}\dot{W}_{l+1},$$

then

$$\dot{D}_l = \frac{d}{dt}\left[W_{l+1}^T W_{l+1} - W_l W_l^\mathsf{T}\right] = \left[W_{l+1}^\mathsf{T}\dot{W}_{l+1} - \dot{W}_l W_l^\mathsf{T}\right] + \left[W_{l+1}^\mathsf{T}\dot{W}_{l+1} - \dot{W}_l W_l^\mathsf{T}\right]^\mathsf{T} = 0.$$

Therefore, $D_l(t) = D_l(0)$. □

*Proof of Theorem 4.1.* From the ZAS initialization, $D_l(t) = D_l(0) = 0$, $l = 1, \ldots, L-2$ and $D_{L-1}(t) = D_{L-1}(0) = -I$, i.e.,

$$W_l W_l^\mathsf{T} = W_{l+1}^\mathsf{T} W_{l+1}, \quad l = 1, \ldots, L-2,$$
$$W_{L-1} W_{L-1}^\mathsf{T} = I + W_L^\mathsf{T} W_L.$$

So we have

$$
\begin{aligned}
W_{L-1:1} W_{L-1:1}^\mathsf{T} &= W_{L-1:2} W_1 W_1^\mathsf{T} W_{L-1:2}^\mathsf{T} = W_{L-1:2} W_2^\mathsf{T} W_2 W_{L-1:2}^\mathsf{T} \\
&= W_{L-1:3}(W_2 W_2^\mathsf{T})^2 W_{L-1:3}^\mathsf{T} \\
&= \cdots \\
&= (W_{L-1} W_{L-1}^\mathsf{T})^{L-1} \\
&= (I + W_L^\mathsf{T} W_L)^{L-1},
\end{aligned}
$$

and

$$
\begin{aligned}
\|\nabla_L \mathcal{R}\|_F^2 &= \left\|(W_{L:1} - \Phi)W_{L-1:1}^\mathsf{T}\right\|_F^2 \\
&\geq \sigma_{\min}^2(W_{L-1:1})\|W_{L:1} - \Phi\|_F^2 = \lambda_{\min}(W_{L-1:1} W_{L-1:1}^\mathsf{T}) \cdot 2\mathcal{R} \\
&= \lambda_{\min}\left((I + W_L^\mathsf{T} W_L)^{L-1}\right) \cdot 2\mathcal{R} \geq 2\mathcal{R}. \tag{4.4}
\end{aligned}
$$

Then

$$\dot{\mathcal{R}}(t) = \sum_{l=1}^L \mathrm{tr}\left(\nabla_l^\mathsf{T}\mathcal{R}(t)\dot{W}_l(t)\right) = -\sum_{l=1}^L \|\nabla_l \mathcal{R}\|_F^2 \leq -\|\nabla_L \mathcal{R}\|_F^2 \leq -2\mathcal{R}.$$

Therefore, $\mathcal{R}(t) \leq e^{-2t}\mathcal{R}(0)$. □

*Remark.* (1). For rectangular weight matrices $W_l \in \mathbb{R}^{d_l \times d_{l-1}}$, if $d_l \geq d_0 = d_{\boldsymbol{x}}$, $l = 1, \ldots, L-1$, we can always ignore the redundant nodes by initializing $W_L = 0$ and $W_l = \begin{bmatrix} I_{d_0} & 0 \\ 0 & 0 \end{bmatrix}$, then the proof of Theorem 4.1 still holds. (2). For the general square loss $\tilde{\mathcal{R}}$ in (2.2) with un-whitened data $X$, if $\lambda_X \overset{\text{def}}{=} \lambda_{\min}(X^\mathsf{T} X) > 0$, following the similar proof, we will have $\|\nabla_L \tilde{\mathcal{R}}\|_F^2 \geq 2\lambda_X \tilde{\mathcal{R}}$, and $\tilde{\mathcal{R}}(t) \leq e^{-2\lambda_X t}\tilde{\mathcal{R}}(0)$.

## 4.2 Discrete-time gradient descent

Now we consider the discrete-time gradient descent (2.4). The main theorem is stated below.

**Theorem 4.3** (Discrete gradient descent). *For deep linear network (2.3) with the zero-asymmetric initialization (3.1) and discrete-time gradient descent (2.4), if the learning rate satisfies*

$$\eta \leq \min\left\{ \left(4L^3\phi^6\right)^{-1}, \left(144L^2\phi^4\right)^{-1} \right\}$$

*where $\phi = \max\left\{2\|\Phi\|_F, 3L^{-1/2}, 1\right\}$, then we have linear convergence*

$$\mathcal{R}(t) \leq \left(1 - \frac{\eta}{2}\right)^t \mathcal{R}(0), \quad t = 0, 1, 2, \ldots \tag{4.5}$$

Since the learning rate $\eta = O\left(L^{-3}\right)$, the theorem indicates that gradient descent can achieve $\mathcal{R}(t) \leq \varepsilon$ in $O\left(L^3 \log(1/\varepsilon)\right)$ iterations.

### 4.2.1 Overview of the proof

The following is the proof sketch, and the detailed proof is deferred to the appendix.

The approach to the discrete-time result is similar to the continuous-time case. However, the matrices defined in (4.3) are not exactly invariant, but change slowly during the training process, which need to be controlled carefully.

First, we propose the following three conditions, and prove that the first condition implies the other two.

**Approximate invariances** For invariant matrices defined in (4.3),

$$\|D_l\|_2 = O\left(L^{-3}\right), \; l = 1, \ldots, L-2, \quad \text{and} \quad \|I + D_{L-1}\|_2 = O\left(L^{-2}\right). \tag{4.6}$$

**Weight bounds** For weight matrices $W_l$,

$$\|W_l\|_2 = 1 + O\left(\frac{\log L}{L}\right), \; l = 1, \ldots, L-1, \quad \text{and} \quad \|W_{L-1}\| = O\left(L^{-1/2}\right). \tag{4.7}$$

**Gradient bound** The gradient of the last layer

$$\|\nabla_L \mathcal{R}\|_F^2 \geq \mathcal{R}. \tag{4.8}$$

**Lemma 4.4.** *The approximate invariances condition (4.6) implies the weight bounds (4.7) and the gradient bound (4.8).*

Second, to show that (4.6)–(4.8) always holds during the training process, we need to estimate the change of invariant matrix $D_l(t+1) - D_l(t)$ and the decrease of loss $\mathcal{R}(t+1) - \mathcal{R}(t)$ in one step.

**Lemma 4.5.** *If the weight bounds (4.7) hold at iteration $t$, then the change of invariant matrices after one-step update with learning rate $\eta$ satisfies*

$$\|D_l(t+1) - D_l(t)\|_2 = O\left(\eta^2\right)\mathcal{R}(t), \; l = 1, \ldots, L-2,$$
$$\|D_{L-1}(t+1) - D_{L-1}(t)\|_2 = O\left(\eta^2 L\right)\mathcal{R}(t). \tag{4.9}$$

**Lemma 4.6.** *If the weight bounds (4.7) and the gradient bound (4.8) hold, and the learning rate $\eta = O\left(L^{-2}\right)$, then the loss function*

$$\mathcal{R}(t+1) \leq \left(1 - \frac{\eta}{2}\right)\mathcal{R}(t). \tag{4.10}$$

With the three lemmas above, we are now ready to prove Theorem 4.3.

*Proof of Theorem 4.3 (informal).* We do induction on (4.5) and (4.6). Assume that they hold for $0, 1, \ldots, t$. From the three lemmas above, (4.7)–(4.10) also hold for $0, 1, \ldots, t$. So the loss function

$$\mathcal{R}(t+1) \leq \left(1 - \frac{\eta}{2}\right)\mathcal{R}(t) \leq \left(1 - \frac{\eta}{2}\right)^{t+1}\mathcal{R}(0),$$

i.e., (4.5) holds for $t + 1$. Now we have

$$\sum_{s=0}^{t} \mathcal{R}(s) \leq \mathcal{R}(0) \sum_{s=0}^{t} \left(1 - \frac{\eta}{2}\right)^s \leq \frac{2}{\eta} \mathcal{R}(0) = O\left(\eta^{-1}\right).$$

Recall that the invariant matrices $D_l(0) = 0$, $l = 1, \ldots, L - 2$ and $I + D_{L-1}(0) = 0$ at the initialization, and $\eta = O\left(L^{-3}\right)$. From (4.9),

$$\|D_l(t+1)\|_2 \leq \sum_{s=0}^{t} \|D_l(s+1) - D_l(s)\|_2 = O(\eta^2) \sum_{s=0}^{t} \mathcal{R}(s) = O(\eta) = O\left(L^{-3}\right).$$

for $l = 1, \ldots, L - 2$. Similarly, $\|I + D_{L-1}(t+1)\|_2 \leq O(\eta L) = O\left(L^{-2}\right)$, i.e., (4.6) holds for $t + 1$. Then we complete the induction. $\qquad\square$

*Remark.* Following the proof sketch, we can actually prove Theorem 4.3 under "near-ZAS" initialization with perturbation: $W_l(0) \sim \mathcal{N}\left(I, \sigma^2\right)$, $l = 1, \ldots, L - 1$ and $W_L(0) \sim \mathcal{N}\left(0, \sigma^2\right)$, where $\sigma$ is sufficiently small such that the approximate invariances condition (4.6) holds at the initialization. Note that the constants hidden in $O(\cdot)$ may depend on the target matrix $\Phi$.

## 5   Numerical experiments

### 5.1   Dependence on the depth

Theorem 4.3 theoretically shows that the number of iterations required for convergence is at most $O\left(L^3\right)$, which holds for any target matrix in $\mathbb{R}^{d \times d}$. The first experiment examines how this depth dependence behaves in practice.

In experiments, we generate target matrices in two ways:

- Gaussian random matrix: $\Phi = (\phi_{ij}) \in \mathbb{R}^{d \times d}$ with $\phi_{ij}$ independently drawn from $\mathcal{N}(0, 1)$. Both $d = 2$ and $d = 100$ are considered.
- Negative identity matrix: $\Phi = -I \in \mathbb{R}^{d \times d}$. This target is adopted from [18], which proves that in the case $d = 1$, the number of iteration required for convergence under the Xavier and the near-identity initialization scales exponentially with the depth $L$. Both $d = 1$ and $d = 100$ are considered.

The ZAS initialization (3.1) is applied for linear neural networks with different depth $L$, and we manually tune the optimal learning rate for each $L$. As suggested by Theorem 4.3, we numerically find that the optimal learning rate decrease with $L$.

Figure 2 shows number of iterations required to make the objective $\mathcal{R} \leq \varepsilon = 10^{-10}$. It is clear to see that the number of iterations required roughly scale as $O\left(L^\gamma\right)$, where $\gamma \approx 1/2$ for the negative identity matrix and $\gamma \approx 1$ for the Gaussian random matrices. These scalings are better than the theoretical $\gamma = 3$ in Theorem 4.3, which is a worst case result.

### 5.2   Comparison with near-identity initialization in multi-dimensional cases

The near-identity initialization initializes each layer by

$$W_l = I + U_l, \quad (U_l)_{ij} \sim \mathcal{N}(0, 1/(dL)) \text{ i.i.d.}, \quad l = 1, \ldots, L \qquad (5.1)$$

where $I$ is the identity matrix. Numerically, it was observed in [18] that for multi-dimensional networks ($d = 25$ in the experiments), gradient descent with the initialization (5.1) requires number of iterations to scale only polynomially with the depth, instead of exponentially. Here we compare it with the ZAS initialization by fitting negative identity matrix with 6-layer linear networks. The learning rate $\eta = 0.01$ for both initialization.

Figure 3 shows the dynamics trajectories for both initializations. It strongly suggests that the ZAS initialization is more efficient than the near-identity initialization (5.1). Gradient descent with the near-identity initialization is attracted to a saddle region, spends a long time escaping that region, and then converges fast to a global minimum. As a comparison, gradient descent with ZAS initialization does not encounter any saddle region during the whole optimization process.

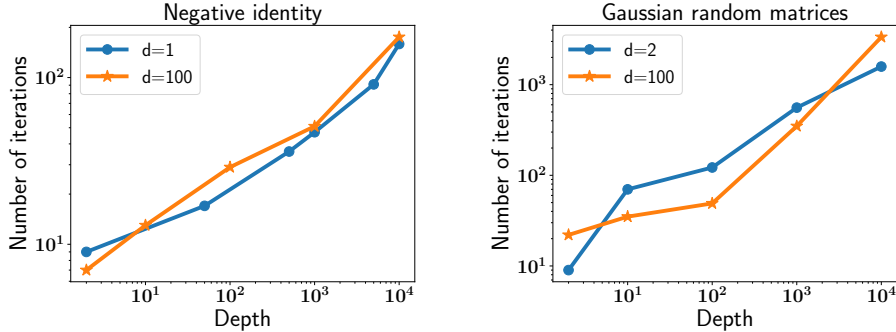

Figure 2: Number of iterations required for the ZAS initialization to reach an $\varepsilon$-optimal solution where $\varepsilon = 10^{-10}$. Two type of target matrices, negative identity and Gaussian random matrices are considered. It is shown that the number of iterations required scales polynomially with the network depth.

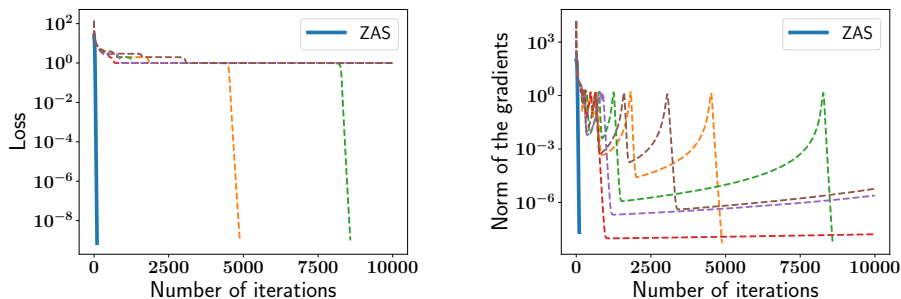

Figure 3: Comparison between the ZAS and the near-identity initialization. The 5 dashed lines correspond to the multiple runs of gradient descent with the near-identity initialization. It is shown that GD with the near-identity successfully escape the saddle region only 2 of 5 times in the given number of iterations, while the ZAS does not suffer from the attraction of saddle point at all.

## 6 An extension to nonlinear Residual networks

Consider the following residual network $f : \mathbb{R}^d \to \mathbb{R}^{d'}$:

$$
\begin{aligned}
\boldsymbol{z}_0 &= V_0 \boldsymbol{x}, \\
\boldsymbol{z}_l &= \boldsymbol{z}_{l-1} + U_l \sigma(V_l \boldsymbol{z}_{l-1}), \quad l = 1, \dots, L, \\
f(\boldsymbol{x}) &= U_{L+1} \boldsymbol{z}_L,
\end{aligned}
\tag{6.1}
$$

where $V_0 \in \mathbb{R}^{D \times d}$, $U_l \in \mathbb{R}^{D \times m}$, $V_l \in \mathbb{R}^{m \times D}$ and $U_{L+1} \in \mathbb{R}^{d' \times D}$; $d$ is the input dimension, $d'$ is the output dimension, $m$ is the width of the residual blocks and $D$ is the width of skip connections.

For the nonlinear residual network (6.1), we propose the following *modified ZAS (mZAS) initialization*:

$$
\begin{aligned}
U_l &= 0, \quad l = 1, 2, \dots, L+1, \\
(V_l)_{ij} &\sim \mathcal{N}(0, 1/D) \text{ i.i.d.}, \quad l = 0, 1, \dots, L.
\end{aligned}
\tag{6.2}
$$

We test two types of initialization: (1) standard Xavier initialization; (2) mZAS initialization (6.2). The experiments are conducted on Fashion-MNIST [20], where we select 1000 training samples forming the new training set to speed up the computation. Depth $L = 100, 200, 2000, 10000$ are tested, and the learning rate for each depth is tuned to the achieve the fastest convergence. The results are displayed in Figure 4.

It is shown that mZAS initialization always outperforms the Xavier initialization. Moreover, gradient descent with mZAS initialization is even able to successfully optimize a 10000-layer residual network. It clearly demonstrates that the ZAS-type initialization can be helpful for optimizing deep nonlinear residual networks.

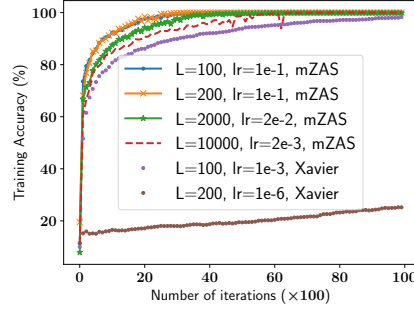

Figure 4: For the nonlinear residual network and Fashion-MNIST dataset, the mZAS initialization outperforms the Xavier initialization. The latter blow up for depth $L = 2000, 10000$. The learning rates are tuned to achieve the fastest convergence.

# 7 Conclusion

In this paper we propose the ZAS initialization for deep linear residual network, under which gradient descent converges to global minima for arbitrary target matrices with linear rate. Moreover, the rate only scales polynomially with the network depth. Numerical experiments show that the ZAS initialization indeed avoids the attraction of saddle points, comparing to the near-identity initialization. This type of initialization may be extended to the analysis of deep nonlinear residual networks, which we leave as future work.

**Acknowledgments**

We are grateful to Prof. Weinan E for helpful discussions, and the anonymous reviewers for valuable comments and suggestions. This work is supported in part by a gift to Princeton University from iFlytek and the ONR grant N00014-13-1-0338.

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
