[Supplementary Material · deep_linear_nips_appendix.pdf]

# A  Proof of the discrete-time gradient descent

## A.1  Invariant matrices

The zero-asymmetry initialization (3.1) gives $D_l(0) = 0$, $l = 1, \ldots, L - 2$ and $I + D_{L-1}(0) = 0$. Lemma 4.2 proved that $D_l$'s are indeed invariances in continuous gradient descent, and then the gradient $\|\nabla_L \mathcal{R}\|_F^2$ can be lower bounded by the current loss $\mathcal{R}$ (4.4). Here we will show that $\|\nabla_L \mathcal{R}\|_F^2 \geq \mathcal{R}$ still holds if $D_l$'s are only *approximately invariant*, i.e., $D_l$, $l = 1, \ldots, L - 2$ and $I + D_{L-1}$ are close to 0.

**Lemma A.1.** *Assume that the weight matrices* $\|W_l\|_2 \leq \alpha$, $l = 1, \ldots, L - 1$ *and the invariant matrices* $\|D_l\|_2 \leq \delta$, $l = 1, \ldots, L - 2$, *then*

$$\left\| W_{L-1:1} W_{L-1:1}^\mathsf{T} - \left( W_{L-1} W_{L-1}^\mathsf{T} \right)^{L-1} \right\|_2 \leq \frac{1}{2} L^2 \alpha^{2(L-2)} \delta. \tag{A.1}$$

*Proof.* We will proof the following statement by induction

$$\left\| W_{l:1} W_{l:1}^\mathsf{T} - \left( W_l W_l^\mathsf{T} \right)^l \right\|_2 \leq \frac{l(l-1)}{2} \alpha^{2(l-1)} \delta, \quad l = 1, \ldots, L - 1. \tag{A.2}$$

The statement holds for $l = 1$ obviously. Assume that the statement holds for $l$, now consider $l + 1$,

$$\left\| W_{l+1:1} W_{l+1:1}^\mathsf{T} - \left( W_{l+1} W_{l+1}^\mathsf{T} \right)^{l+1} \right\|_2$$
$$= \left\| W_{l+1} \left[ W_{l:1} W_{l:1}^\mathsf{T} - \left( W_l W_l^\mathsf{T} \right)^l + \left( W_l W_l^\mathsf{T} \right)^l - \left( W_{l+1}^\mathsf{T} W_{l+1} \right)^l \right] W_{l+1}^\mathsf{T} \right\|_2$$
$$\leq \|W_{l+1}\|_2 \left[ \left\| W_{l:1} W_{l:1}^\mathsf{T} - \left( W_l W_l^\mathsf{T} \right)^l \right\|_2 + \left\| \left( W_l W_l^\mathsf{T} \right)^l - \left( W_{l+1}^\mathsf{T} W_{l+1} \right)^l \right\|_2 \right] \|W_{l+1}^\mathsf{T}\|_2$$
$$\leq \alpha^2 \left[ \frac{l(l-1)}{2} \alpha^{2(l-1)} \delta + \left\| \left( W_l W_l^\mathsf{T} \right)^l - \left( W_{l+1}^\mathsf{T} W_{l+1} \right)^l \right\|_2 \right],$$

and

$$\left\| \left( W_l W_l^\mathsf{T} \right)^l - \left( W_{l+1}^\mathsf{T} W_{l+1} \right)^l \right\|_2$$
$$= \left\| \sum_{k=0}^{l-1} \left( W_l W_l^\mathsf{T} \right)^{l-1-k} \left( W_l W_l^\mathsf{T} - W_{l+1}^\mathsf{T} W_{l+1} \right) \left( W_{l+1}^\mathsf{T} W_{l+1} \right)^k \right\|_2$$
$$\leq \sum_{k=0}^{l-1} \left\| W_l W_l^\mathsf{T} \right\|_2^{l-1-k} \left\| W_l W_l^\mathsf{T} - W_{l+1}^\mathsf{T} W_{l+1} \right\|_2 \left\| W_{l+1}^\mathsf{T} W_{l+1} \right\|_2^k$$
$$\leq \sum_{k=0}^{l-1} \alpha^{2(l-1-k)} \delta \alpha^{2k} = l \alpha^{2(l-1)} \delta,$$

thus

$$\left\| W_{l+1:1} W_{l+1:1}^\mathsf{T} - \left( W_{l+1} W_{l+1}^\mathsf{T} \right)^{l+1} \right\|_2 \leq \alpha^2 \left[ \frac{l(l-1)}{2} \alpha^{2(l-1)} \delta + l \alpha^{2(l-1)} \delta \right] = \frac{l(l+1)}{2} \alpha^{2l} \delta.$$

So the statement (A.2) also holds for $l + 1$, and we complete the proof of the lemma. $\quad\square$

**Lemma A.2.** *Assume that the weight matrices* $\|W_l\|_2 \leq \alpha$, $l = 1, \ldots, L - 1$, *where* $1 \leq \alpha^{2(L-1)} < L\phi^2$ *for some* $\phi > 0$; *assume that the invariant matrices* $\|D_l\|_2 \leq \delta$, $l = 1, \ldots, L - 2$ *and* $\|I + D_{L-1}\|_2 \leq \varepsilon$, *where* $\delta \leq \left( 2L^3 \phi^2 \right)^{-1}$ *and* $\varepsilon \leq \left( 4L^2 \right)^{-1}$. *Then* $\|\nabla_L \mathcal{R}\|_F^2 \geq \mathcal{R}$.

*Proof.* From Lemma A.1,

$$\lambda_{\min} \left( W_{L-1:1} W_{L-1:1}^\mathsf{T} \right) \geq \lambda_{\min}^{L-1} \left( W_{L-1} W_{L-1}^\mathsf{T} \right) - \frac{1}{2} L^2 \alpha^{2(L-2)} \delta.$$

Since $\left\| I + W_L^\mathsf{T} W_L - W_{L-1} W_{L-1}^\mathsf{T} \right\|_2 \leq \varepsilon$,

$$\lambda_{\min}(W_{L-1} W_{L-1}^\mathsf{T}) \geq \lambda_{\min} \left( I + W_L^\mathsf{T} W_L \right) - \varepsilon \geq 1 - \varepsilon.$$

Similar to (4.4), we have

$$\|\nabla_L \mathcal{R}\|_F^2 \geq 2\lambda_{\min}\left(W_{L-1:1}W_{L-1:1}^\mathsf{T}\right)\mathcal{R}$$

$$\geq 2\left[(1-\varepsilon)^{L-1} - \frac{1}{2}L^2\alpha^{2(L-2)}\delta\right]\mathcal{R} \geq 2\left[1 - \frac{L-1}{4L^2} - \frac{1}{2}L^2 \cdot L\phi^2 \cdot \frac{1}{2L^3\phi^2}\right]\mathcal{R} \geq \mathcal{R}.$$

$\square$

In addition, if $D_l$'s are approximately invariant, we can bound the weights $\|W_l\|_2$.

**Lemma A.3.** *Let* $\alpha = \max_{1 \leq l \leq L-1}\|W_l\|_2 \vee 1$, $\beta = \|W_L\|_2$ *and* $\phi = \max\left\{\|W_{L:1}\|_2, \frac{e}{\sqrt{L}}, 1\right\}$. *Assume that the invariant matrices* $\|D_l\|_2 \leq \delta$, $l = 1, \ldots, L-2$ *and* $\|I + D_{L-1}\|_2 \leq \varepsilon$, *where* $\delta \leq \left(2L^3\phi^2\right)^{-1}$ *and* $\varepsilon \leq \left(4L^2\right)^{-1}$. *Then*

$$\alpha^{2(L-1)} < L\phi^2, \quad \alpha^{2(L-1)}\beta^2 < 2\phi^2. \tag{A.3}$$

*Proof.* We first use the invariant matrices to bound the difference between $\|W_l\|_2$. Since

$$\|I + D_{L-1}\|_2 = \left\|I + W_L^\mathsf{T}W_L - W_{L-1}W_{L-1}^\mathsf{T}\right\|_2$$
$$\geq \left|\left\|I + W_L^\mathsf{T}W_L\right\|_2 - \left\|W_{L-1}W_{L-1}^\mathsf{T}\right\|_2\right| = \left|1 + \|W_L\|_2^2 - \|W_{L-1}\|_2^2\right|,$$

we have $\left|1 + \beta^2 - \|W_{L-1}\|_2^2\right| \leq \varepsilon$. In addition,

$$\|D_l\|_2 = \left\|W_{l+1}^\mathsf{T}W_{l+1} - W_lW_l^\mathsf{T}\right\|_2 \geq \left|\left\|W_{l+1}^\mathsf{T}W_{l+1}\right\|_2 - \left\|W_lW_l^\mathsf{T}\right\|_2\right| = \left|\|W_{l+1}\|_2^2 - \|W_l\|_2^2\right|$$

for $l = 1, \ldots, L-2$, then $\left|1 + \beta^2 - \|W_l\|_2^2\right| \leq \varepsilon + (L-l-1)\delta$, thus $\left|1 + \beta^2 - \alpha^2\right| \leq \varepsilon + (L-2)\delta$. From Lemma A.1,

$$W_{L:1}W_{L:1}^\mathsf{T} = W_L\left[W_{L-1:1}W_{L-1:1}^\mathsf{T}\right]W_L^\mathsf{T}$$
$$\succeq W_L\left[\left(W_{L-1}W_{L-1}^\mathsf{T}\right)^{L-1} - \frac{1}{2}\alpha^{2(L-2)}L^2\delta I\right]W_L^\mathsf{T}$$
$$\succeq W_L\left[\left(I + W_L^\mathsf{T}W_L - \delta I\right)^{L-1} - \frac{1}{2}\alpha^{2(L-2)}L^2\delta I\right]W_L^\mathsf{T},$$

where $A \succeq B$ means the matrix $A - B$ is positive semi-definite. So

$$\left\|W_{L:1}W_{L:1}^\mathsf{T}\right\|_2 \geq \left\|W_L\left[\left(I + W_L^\mathsf{T}W_L - \delta I\right)^{L-1} - \frac{1}{2}\alpha^{2(L-2)}L^2\delta I\right]W_L^\mathsf{T}\right\|_2$$
$$= \beta^2\left[\left(1 + \beta^2 - \varepsilon\right)^{L-1} - \frac{1}{2}\alpha^{2(L-2)}L^2\delta\right]$$
$$\geq \beta^2\left[\left(\alpha^2 - 2\varepsilon - (L-2)\delta\right)^{L-1} - \frac{1}{2}\alpha^{2(L-2)}L^2\delta\right]$$
$$\geq \beta^2\left[\alpha^{2(L-1)} - (L-1)(2\varepsilon + (L-2)\delta) - \frac{1}{2}\alpha^{2(L-2)}L^2\delta\right],$$
$$\geq \beta^2\left[\alpha^{2(L-1)} - \frac{(L-1)^2}{L^3} - \frac{1}{4L}\alpha^{2(L-2)}\right]$$
$$\geq \frac{1}{2}\alpha^{2(L-1)}\beta^2$$

since $\delta \leq \left(2L^3\right)^{-1}$ and $\varepsilon \leq \left(4L^2\right)^{-1}$. Therefore, $\alpha^{2(L-1)}\beta^2 \leq 2\left\|W_{L:1}W_{L:1}^\mathsf{T}\right\|_2 \leq 2\phi^2$.

Finally, assume that $\alpha^{2(L-1)} \geq L\phi^2$, then

$$\alpha^2 \geq \left(L\phi^2\right)^{1/(L-1)} = \exp\left[\frac{\log\left(L\phi^2\right)}{L-1}\right] > 1 + \frac{\log\left(L\phi^2\right)}{L-1},$$
$$\beta^2 \geq \frac{\log\left(L\phi^2\right)}{L-1} - \varepsilon - (L-2)\delta \geq \frac{2}{L-1} - \frac{1}{4L^2} - \frac{L-2}{2L^3} > \frac{2}{L},$$

where $\log(L\phi^2) \geq 2$ comes from $\phi \geq \frac{e}{\sqrt{L}}$. Thus

$$\left\|W_{L:1}W_{L:1}^\mathsf{T}\right\|_2 \geq \frac{1}{2}\alpha^{2(L-1)}\beta^2 > \frac{1}{2} \cdot L\phi^2 \cdot \frac{2}{L} = \phi^2,$$

which is a contradiction! Therefore $\alpha^{2(L-1)} < L\phi^2$, and we complete the proof of the lemma. $\square$

## A.2 One-step analysis

We denote the one-step update as

$$W_l^+ = W_l - \eta\nabla_l\mathcal{R}, \quad l = 1,\ldots,L.$$

In this section, we always denote $A^+$ as the value of a variable $A$ after one-step update, for example $\mathcal{R}^+$, $W_{l_2:l_1}^+$ and $D_l^+$. We will estimate the change of invariant matrix $D_l^+ - D_l$ and the change of loss $\mathcal{R}^+ - \mathcal{R}$ in one step.

**Lemma A.4.** *Assume that $\|W_l\|_2 \leq \alpha$, $l = 1,\ldots,L-1$ and $\|W_L\|_2 \leq \beta$, where $1 \leq \alpha^{2(L-1)} < L\phi^2$ and $\alpha^{2(L-1)}\beta^2 < 2\phi^2$ for some $\phi > 0$. Then*

$$\|\nabla_l\mathcal{R}\|_F^2 \leq 4\phi^2\mathcal{R}, \quad l = 1,\ldots,L-1,$$
$$\|\nabla_L\mathcal{R}\|_F^2 \leq 2L\phi^2\mathcal{R}.$$

*Proof.* For $l = 1,\ldots,L-1$,

$$\|\nabla_l\mathcal{R}\|_F = \left\|W_{L:(l+1)}^\mathsf{T}(W_{L:1} - \Phi)W_{(l-1):1}^\mathsf{T}\right\|_F \leq \left\|W_{L:(l+1)}\right\|_2 \left\|W_{L:1} - \Phi\right\|_F \left\|W_{(l-1):1}\right\|_2$$
$$\leq \alpha^{L-2}\beta\sqrt{2\mathcal{R}} \leq 2\phi\sqrt{\mathcal{R}}.$$

And similarly, $\|\nabla_L\mathcal{R}\|_F \leq \alpha^{L-1}\sqrt{2\mathcal{R}} \leq \phi\sqrt{2L\mathcal{R}}$. $\square$

**Lemma A.5.** *Under the same conditions as Lemma A.4, the change of invariant matrices under one-step update satisfies*

$$\|D_l^+ - D_l\|_2 \leq 8\eta^2\phi^2\mathcal{R}, \quad l = 1,\ldots,L-2,$$
$$\|D_{L-1}^+ - D_{L-1}\|_2 \leq 2\eta^2(L+2)\phi^2\mathcal{R}.$$

*Proof.* Recall the invariance condition

$$\nabla_l\mathcal{R}W_l^\mathsf{T} = W_{L:(l+1)}^\mathsf{T}(W_{L:1} - \Phi)W_{l:1}^\mathsf{T} = W_{l+1}^\mathsf{T}\nabla_{l+1}\mathcal{R},$$

we have

$$\begin{aligned}
D_l^+ &= (W_{l+1}^+)^\mathsf{T}W_{l+1}^+ - W_l(W_l^+)^\mathsf{T}\\
&= (W_{l+1} - \eta\nabla_{l+1}\mathcal{R})^\mathsf{T}(W_{l+1} - \eta\nabla_{l+1}\mathcal{R}) - (W_l - \eta\nabla_l\mathcal{R})(W_l - \eta\nabla_l\mathcal{R})^\mathsf{T}\\
&= W_{l+1}^\mathsf{T}W_{l+1} - W_lW_l^\mathsf{T}\\
&\quad - \eta\left[W_{l+1}^\mathsf{T}\nabla_{l+1}\mathcal{R} - \nabla_l\mathcal{R}W_l^\mathsf{T} + \nabla_{l+1}^\mathsf{T}\mathcal{R}W_{l+1} - W_l\nabla_l^\mathsf{T}\mathcal{R}\right]\\
&\quad + \eta^2\left[\nabla_{l+1}^\mathsf{T}\mathcal{R}\nabla_{l+1}\mathcal{R} + \nabla_l\mathcal{R}\nabla_l^\mathsf{T}\mathcal{R}\right]\\
&= D_l + \eta^2\left[\nabla_{l+1}^\mathsf{T}\mathcal{R}\nabla_{l+1}\mathcal{R} + \nabla_l\mathcal{R}\nabla_l^\mathsf{T}\mathcal{R}\right].
\end{aligned}$$

Combining with Lemma A.4, we can complete the proof. $\square$

**Lemma A.6.** *Under the same conditions as Lemma A.2, for learning rate*

$$\eta \leq \min\left\{\frac{1}{64L^2\phi^3\sqrt{\mathcal{R}}}, \frac{1}{144L^2\phi^4}\right\},$$

*the decrease of the loss function in one-step update satisfies*

$$\mathcal{R}^+ \leq \left(1 - \frac{\eta}{2}\right)\mathcal{R}.$$

*Proof.* First we expand $W_{L:1}^+$ as a polynomials of $\eta$:

$$W_{L:1}^+ = \prod_{l=1}^{L} (W_l - \eta\nabla_l\mathcal{R}) = A_0 + \eta A_1 + \eta^2 A_2 + \cdots + \eta^L A_L,$$

where the coefficients $A_k \in \mathbb{R}^{d\times d}$. Obviously $A_0 = W_{L:1}$.

$$\begin{aligned}
\mathcal{R}^+ - \mathcal{R} &= \frac{1}{2}\left[\left\|W_{L:1}^+ - \Phi\right\|_F^2 - \left\|W_{L:1} - \Phi\right\|_F^2\right] \\
&= \frac{1}{2}\left(W_{L:1}^+ - W_{L:1}\right) : \left(W_{L:1}^+ + W_{L:1} - 2\Phi\right) \\
&= \frac{1}{2}\left(W_{L:1}^+ - W_{L:1}\right) : \left(2\left(W_{L:1} - \Phi\right) + \left(W_{L:1}^+ - W_{L:1}\right)\right) \\
&= \left(W_{L:1}^+ - W_{L:1}\right) : \left(W_{L:1} - \Phi\right) + \frac{1}{2}\left\|W_{L:1}^+ - W_{L:1}\right\|_F^2,
\end{aligned}$$

where $A : B = \sum_{i,j} A_{ij}B_{ij}$. We can write

$$\mathcal{R}^+ - \mathcal{R} = I_1 + I_2 + I_3,$$

where

$$I_1 = \eta A_1 : (W_{L:1} - \Phi), \quad I_2 = \sum_{k=2}^{L} \eta^k A_k : (W_{L:1} - \Phi), \quad I_3 = \frac{1}{2}\left\|\sum_{k=1}^{L}\eta^k A_k\right\|_F^2.$$

For $I_1$, we have

$$\begin{aligned}
I_1 = A_1 : (W_{L:1} - \Phi) &= -\eta\sum_{l=1}^{L}(W_{L:l+1}\nabla_l\mathcal{R}W_{l-1:1}) : (W_{L:1} - \Phi) \\
&= -\eta\sum_{l=1}^{L}\nabla_l\mathcal{R} : \left[W_{L:l+1}^{\mathsf{T}}\left(W_{L:1} - \Phi\right)W_{l-1:1}^{\mathsf{T}}\right] = -\eta\sum_{l=1}^{L}\|\nabla_l\mathcal{R}\|_F^2.
\end{aligned}$$

From Lemma A.2,

$$I_1 \le -\eta\|\nabla_L\mathcal{R}\|_F^2 \le -\eta\mathcal{R}.$$

For $I_2$ and $I_3$, we further expand $W_{L-1:1}^+$ as

$$W_{L-1:1}^+ = \prod_{l=1}^{L-1}(W_l - \eta\nabla_l\mathcal{R}) = B_0 + \eta B_1 + \eta^2 B_2 + \cdots + \eta^{L-1}B_{L-1}.$$

From Lemma A.4, $\|\nabla_l\mathcal{R}\|_F \le \gamma = 2\phi\sqrt{\mathcal{R}}$, $l = 1, \ldots, L-1$, then for $k \ge 1$,

$$\|B_k\|_F \le \binom{L-1}{k}\alpha^{L-1-k}\left(2\phi\sqrt{\mathcal{R}}\right)^k.$$

We use the following inequalities for $0 \le y \le x/L^2$:

$$(x+y)^L \le 2x^L, \quad (x+y)^L \le x^L + 2Lx^{L-1}y, \quad (x+y)^L \le x^L + Lx^{L-1}y + L^2x^{L-2}y^2.$$

Since $2\eta\phi\sqrt{\mathcal{R}} \le \alpha/L^2$,

$$\left\|\sum_{k=0}^{L-1}\eta^k B_k\right\|_2 \le \left(\alpha + 2\eta\phi\sqrt{\mathcal{R}}\right)^{L-1} \le 2\alpha^{L-1},$$

$$\left\|\sum_{k=1}^{L-1}\eta^k B_k\right\|_F \le \left(\alpha + 2\eta\phi\sqrt{\mathcal{R}}\right)^{L-1} - \alpha^{L-1} \le 2L\alpha^{L-2}\cdot 2\eta\phi\sqrt{\mathcal{R}} = 4\eta L\alpha^{L-2}\phi\sqrt{\mathcal{R}},$$

$$\begin{aligned}
\left\|\sum_{k=2}^{L-1}\eta^k B_k\right\|_F &\le \left(\alpha + 2\eta\phi\sqrt{\mathcal{R}}\right)^{L-1} - \alpha^{L-1} - (L-1)\alpha^{L-2}\cdot 2\eta\phi\sqrt{\mathcal{R}} \\
&\le L^2\alpha^{L-3}\left(2\eta\phi\sqrt{\mathcal{R}}\right)^2 = 4\eta^2 L^2\alpha^{L-3}\phi^2\mathcal{R}.
\end{aligned}$$

Notice that $A_k = W_L B_k - \nabla_L \mathcal{R} B_{k-1}$, $k = 1, \ldots, L$ where $\|W_L\|_2 \leq \beta$ and $\|\nabla_L \mathcal{R}\|_F \leq \alpha^{L-1}\sqrt{2\mathcal{R}}$, then

$$
\begin{aligned}
\left\|\sum_{k=1}^{L} \eta^k A_k\right\|_F &\leq \|W_L\|_2 \left\|\sum_{k=1}^{L} \eta^k B_k\right\|_F + \eta\|\nabla_L \mathcal{R}\|_F \left\|\sum_{k=0}^{L} \eta^k B_k\right\|_2 \\
&\leq \beta \cdot 4\eta L \alpha^{L-2}\phi\sqrt{\mathcal{R}} + \eta\alpha^{L-1}\sqrt{2\mathcal{R}} \cdot 2\alpha^{L-1} \\
&\leq 4\eta L \phi^2 \sqrt{2\mathcal{R}} + 2\eta L \alpha^2 \phi^2 \sqrt{2\mathcal{R}} \\
&= 6\eta L \phi^2 \sqrt{2\mathcal{R}},
\end{aligned}
$$

$$
\begin{aligned}
\left\|\sum_{k=2}^{L} \eta^k A_k\right\|_F &\leq \|W_L\|_2 \left\|\sum_{k=2}^{L} \eta^k B_k\right\|_F + \eta\|\nabla_L \mathcal{R}\|_F \left\|\sum_{k=1}^{L} \eta^k B_k\right\|_F \\
&\leq \beta \cdot 4\eta^2 L^2 \alpha^{L-3}\phi^2\mathcal{R} + \eta\alpha^{L-1}\sqrt{2\mathcal{R}} \cdot 4\eta L\alpha^{L-2}\phi\sqrt{\mathcal{R}} \\
&\leq 4\sqrt{2}\eta^2 L^2 \phi^3 \mathcal{R} + 4\sqrt{2}\eta^2 L^2 \phi^3 \mathcal{R} \\
&= 8\sqrt{2}\eta^2 L^2 \phi^3 \mathcal{R}.
\end{aligned}
$$

So

$$
I_2 \leq \left\|\sum_{k=2}^{L} \eta^k A_k\right\|_F \|W_{L:1}(k) - \Phi\|_F \leq 16\eta^2 L^2 \phi^3 \mathcal{R}^{3/2},
$$

$$
I_3 = \frac{1}{2}\left\|\sum_{k=1}^{L} \eta^k A_k\right\|_F^2 \leq 36\eta^2 L^2 \phi^4 \mathcal{R}.
$$

For $\eta \leq \min\left\{\left(64L^2\phi^3\sqrt{\mathcal{R}}\right)^{-1}, \left(144L^2\phi^4\right)^{-1}\right\}$, we have $I_2 \leq \eta\mathcal{R}/4$ and $I_3 \leq \eta\mathcal{R}/4$. Therefore,

$$
\mathcal{R}^+ - \mathcal{R} = I_1 + I_2 + I_3 \leq -\eta\mathcal{R} + \frac{1}{4}\eta\mathcal{R} + \frac{1}{4}\eta\mathcal{R} = -\frac{1}{2}\eta\mathcal{R}.
$$

$\square$

### A.3 Proof of Theorem 4.3

Now we are ready to prove the main Theorem 4.3.

*Proof of Theorem 4.3.* Let $\alpha(t) = \max_{1 \leq l \leq L-1} \|W_l(t)\|_2 \vee 1$, and $\beta(t) = \|W_L(t)\|_2$. We will proof the following two statements by induction:

$$
\alpha^{2(L-1)}(t) < L\phi^2, \quad \alpha^{2(L-1)}(t)\beta^2(t) < 2\phi^2, \tag{A.4}
$$

$$
\mathcal{R}(t) \leq \left(1 - \frac{\eta}{2}\right)^t \mathcal{R}(0). \tag{A.5}
$$

Recall that $\phi = \max\left\{2\|\Phi\|_F, \frac{e}{\sqrt{L}}, 1\right\}$.

The statements hold for $t = 0$ since $\alpha(0) = 1$ and $\beta(0) = 0$. Assume that the statements hold for $0, 1, \ldots, t$, now consider $t + 1$.

From the induction assumption, $\mathcal{R}(t) \leq \mathcal{R}(0) = \phi^2/8$, then $\eta \leq \left(144L^2\phi^4\right)^{-1} < \left(64L^2\phi^3\sqrt{\mathcal{R}(t)}\right)^{-1}$ satisfies the requirement of Lemma A.6. So (A.5) holds for $t + 1$. Furthermore,

$$
\|W_{L:1}(t+1)\|_2 \leq \|W_{L:1}(t+1)\|_F \leq \|\Phi\|_F + \sqrt{2\mathcal{R}(t+1)} \leq \|\Phi\|_F + \sqrt{2\mathcal{R}(0)} \leq \phi.
$$

The invariant matrices $D_l(0) = 0$, $l = 1, \ldots, L-2$ and $I + D_{L-1}(0) = 0$ for initialization. From Lemma A.5, each update

$$
\|D_l(s+1) - D_l(s)\|_2 \leq 8\eta^2 \phi^2 \mathcal{R}(s)
$$

for $l = 1, \ldots, L - 2$ and $s = 0, 1, \ldots, t$. From the induction assumption,

$$\sum_{s=0}^{t} \mathcal{R}(s) \leq \mathcal{R}(0) \sum_{s=0}^{t} \left(1 - \frac{\eta}{2}\right)^s \leq \frac{2}{\eta} \mathcal{R}(0) = \frac{\phi^2}{4\eta},$$

then

$$\|D_l(t+1)\|_2 \leq \sum_{s=0}^{t} \|D_l(s+1) - D_l(s)\|_2 \leq 8\eta^2 \phi^2 \sum_{s=0}^{t} \mathcal{R}(s) \leq 2\eta\phi^4 \leq \frac{1}{2L^3 \phi^2}$$

since $\eta \leq \left(4L^3 \phi^6\right)^{-1}$. Similarly, $\|I + D_{L-1}(t+1)\|_2 \leq 4\eta(L+2)\phi^2 < \left(4L^2\right)^{-1}$. Now from Lemma A.3, the statement (A.4) holds for $t + 1$. Then we complete the induction. $\quad\square$