[Reviews · NeurIPS 2019]

Reviewer 1



Response to authors' feedback: I thank the authors for the rebuttal. My score remains the same. ----------------------------------- The work proposes an initialization for deep linear networks in which the last layer is initialized to 0, and the rest to the identity Id. With this initialization, the networks are shown to converge linearly to zero loss, under conditions (for discrete-time GD) that are different from and perhaps conceptually simpler than previous works. For instance, compared to reference [2] (Arora et al “A convergence analysis of gradient descent for deep linear neural networks”, ICLR 2019), this work removes completely the delta-balanced condition in [2] by showing that this condition actually holds, for most layers, on the GD trajectory (Lemma 4.2 and Eq. (4.6)). While certain elements have already been seen in previous works (e.g. the property in Lemma 4.2 is similar to the delta-balanced condition in [2], or the requirement of zero initialization for the last layer’s weight has been seen in “fixup initialization” of reference [21] in the context of residual networks), I think the proposed initialization as well as the convergence analysis here deserve credits for novelty. In particular, I appreciate the insight that at the region where symmetry breaks, the velocity is bounded away from zero (line 146 and Eq (4.4)), and hence one ought to initialize accordingly. Interestingly, from my understanding, some further extensions of this analysis might say more about the landscape on which GD travels. For example: - Firstly, by symmetry, one can actually choose any one weight matrix to be the zero-initialized one, with the rest initialized to Id. - Secondly, one can modify the ZAS and initialize the last layer as non-zero appropriately (say, 0.1*Id), such that the two identities after line 145 hold. These points further attest some degree of distinction from “fixup initialization” of [21] for this work. - Thirdly, I think instead of initializing to Id, one can initialize to alpha*Id, for alpha>1, and get an improvement on the convergence rate. In particular — if my mental calculation is correct — that should yield, for continuous-time GD: R(t) \leq \exp(-2\alpha^{2(L-1)}t)R(0). Note that this alpha*Id initialization doesn’t change R(0) as compared to Id initialization, since one of the matrices is 0 at initialization. This improvement is significant for larger L. Now we can see that the convergence rate can be made arbitrarily large! The above three points, if true, should show that there might be hidden insights from the analysis of this paper.

Reviewer 2



I've read the author feedback and appreciate the added experiment, which shows the benefit of the proposed ZAS initialization in training very deep ResNets compared with standard init schemes (the authors tried the Xavier init.) For the moment I am raising my score to 6. ------ This paper proves a simple theoretical result for optimizing a deep linear network: when all layers are d-by-d, if we initialize the top layer as 0 and all but the top layer by the identity matrix, then the gradient flow (as well as gradient descent with a polynomially small stepsize) enjoys linear convergence to zero risk on the matrix regression problem. This paper is well presented and the convergence result is particularly neat. However the result is a rather direct consequence of a known alignment property on deep linear networks, and thus might be incremental in its contribution and novelty. Indeed, the proof follows straightforwardly from the known fact (as the authors noted) that the matrices W_{l+1}^T * W_{l+1} - W_l * W_l^T remains constant on the gradient flow path. This paper comes up with a specific design which utilizes the above fact and implies that W_{(L-1):1} stays well-conditioned, and hence a gradient domination condition. Compared with existing alignment-type results on deep linear nets (e.g. Ji and Telgarsky 2019), the present result is a bit lacking in interesting messages for real problems, e.g. when nonlinearity is present.

Reviewer 3



# Positive aspects * The toy model of Figure 1 is insightful and effective at providing an intuition behind the potential advantages of the method. # Post-rebuttal Authors have responded to some of my criticisms, increasing my score to 6. * I checked the proofs in Page 5 and they seem correct to me. # Main criticism The main limitation I see in this paper is that experiments are exclusively done on linear networks. While linear networks might provide a useful tool for the theoretical analysis of initialization and to derive convergence rates, they are not useful machine learning models. In other words, I do not think the development of a new initialization method is useful unless its shown to be competitive on real problems (i.e., beyond deep linear networks). Furthermore, I have concerns with the experiments the authors show even in this unrealistic scenario. The authors only show _one_ run of their experiments where ZAS outperforms the near-identity initialization. It is hence unclear whether this is a reproducible effect or an exception one that only arises in this very specific example that the authors hand-picked. It would be much more convincing in any case to show these curves for multiple runs of the same dataset and not just one hand-picked. # Other comments The paper is rather poorly written. Take for example the description of related work in L17-23. The authors write "the random orthogonal initialization proposed in analyzing the gradient descent ..." are the authors citing a paper named "analyzing the gradient descent ..." ? are they describing the paper?, or in the next line "how the knowledge transfer across tasks in transfer learning" [what is knowledge? transfer-> transfers? furthermore, how is this related to the current paper? Almost every paragraph has some example of poor writing like this, making the paper hard to read. Poor writing is not limited to prose. The mathematical notation is equally sloppy. For example, R is defined in Eq. (2.3) as a function of W_1, ..., W_L. It then appears right below in (2.4), where its instead a function of the iterations t. I guess that R(t) is overloaded to mean R(W_1, ..., W_L), where the W's are taken at the t-th iterate, but I shouldn't have to guess. # Suggestions It seems to be that A : B = tr(A B^T), why not use this more standard notation?

[Author Response · NeurIPS 2019]

We are grateful for all the reviewers' valuable suggestions and questions. We start by showing some additional
experiments for deep nonlinear ResNets. Consider a nonlinear ResNet $f(x;\theta) := w^T z_L$ with $z_L$ recursively defined as

$$z_0 = V_0 x; \quad z_l = z_{l-1} + U_l \sigma(V_l z_{l-1}), \quad l = 1, \dots, L$$

where $V_0 \in \mathbb{R}^{D \times d}, U_l \in \mathbb{R}^{D \times m}, V_l \in \mathbb{R}^{m \times D}$ and $w \in \mathbb{R}^D$. We
test two initializations: (1) standard Xavier initialization; (2) modified
zero-asymmetry(mZAS) initialization : $U_l = 0, w = 0$ and $(V_l)_{i,j} \sim$
$\mathcal{N}(0, 1/D)$. The experiments are conducted on Fashion-MNIST, where we
select 1000 training samples forming the new training set to speed up the
computation. The results are displayed in Figure 1.

We can see that mZAS initialization always outperforms the Xavier ini-
tialization. Moreover, GD with mZAS initialization is able to successfully
optimize a 10000-layer ResNet. It is clearly demonstrated that ZAS-type
initialization can be helpful for optimizing deep nonlinear ResNets. How-
ever, to make this initialization practical for real scenarios such as ImageNet
still requires more efforts, which is beyond the scope of this paper. We will
add a section in the paper to discuss how to adapt it for nonlinear residual
network and provide some preliminary experiments.

Figure 1: The comparison of training curves between two initializations. The learning rate is manually tuned to achieve the best convergence performance. The curves of GD with Xavier initialization for L=2000,10000 are not shown, since they always blow up.

**For Reviewer 2:** **(1)** (for continuous-time GD: $R(t) \leq \exp(-2\alpha^{2(L-1)}t)R(0)$.) Thanks to the reviewer for pointing
out this interesting phenomenon that we did not notice it before. It increases the gradient by $\alpha^{2(L-1)}$ times so the faster
rate holds for continuous-time GD. However, discrete-time GD $R(t) < (1-\alpha^{2(L-1)}\eta/2)^t R(0)$ requires $\eta \lesssim 1/\alpha^{2(L-1)}$,
thus the number of iterations may not exponentially decrease. **(2)** (general case of rectangular weight matrices) This
can be achieved by padding zeros as long as $\min\{d_1, \dots, d_{L-2}\} \geq \min\{d_0, d_{L-1}\}$. Actually, Our analysis works for
a general initialization. Let $m = \min\{d_{L-1}, d_0\}$ and $A = U\Sigma V$ where $U \in \mathbb{R}^{d_{L-1} \times m}, V \in \mathbb{R}^{d_0 \times m}, \Sigma \in \mathbb{R}^{m \times m}$ be
the singular value decomposition of $A$. We can initialize

$$W_L = 0; W_{L-1} \simeq U\Sigma^{1/(L-1)}, W_{L-2} \simeq \Sigma^{1/(L-1)}, \dots, W_2 \simeq \Sigma^{1/(L-1)}, W_1 \simeq \Sigma^{1/(L-1)}V, \tag{2}$$

where the symbol "$\simeq$" stands for equality up to zero-valued paddings. This initialization is similar as the Procedure 1 in
(Arora et al. ICLR2019), but with the top layer to be zero.

**For Reviewer 3:** **(1)** (all layers are $d \times d$) Our proof only relies on the *dynamic invariance* and top layer to be
zero. So the result also holds for the general case, $\min\{d_1, \dots, d_{L-1}\} \geq \min\{d_0, d_L\}$, in which the matrices are
not necessarily square. We will clarify this in the revised version. **(2)** (direct consequence of a known alignment
property...) This known property actually has been widely used in the previous works (Bartlett et al. ICML2018,
Arora et al. ICLR2019, Shamir COLT2019, Du et al. ICML 2019) for analyzing the optimization of linear networks,
but coming up with the right initialization to fully utilize this property is not straightforward. That is why the global
convergence of GD for general linear networks has not been established until this submission. Especially, the picture of
symmetry break behind the ZAS initialization could be useful for analyzing linear networks in other setting, such as
matrix factorization, binary classification, etc.. **(3)** (empirical results on how this specific initialization may help in
practice) We have conducted some experiments, and please refer to the beginning of this rebuttal for the results. **(4)**
(results on deep linear nets (e.g. Ji and Telgarsky, 2019)) Thanks for pointing out this reference and we will add it to the
related work section. This work studies the properties of solutions that the GD converges to, without providing any
convergence rate. The ZAS initialization might help to establish the convergence in their setting.

**For Reviewer 4:** **(1)** (...the development of a new initialization method is useful unless its shown to be competitive
on real problems) We have done some experiments for real problems (given in the beginning of this reresponse).
and the results suggest that the ZAS-type initialization is use-
ful for nonlinear ResNets in practice. However, we want to
stress that the goal of this submission is to provide theoret-
ical understanding of the optimization of deep linear nets.
The ZAS initialization is proposed to obtain a global conver-
gence guarantee of GD for optimizing deep linear nets. **(2)**
(more convincing in any case to show these curves for multiple
runs...) Please see right figure, which shows results of multiple
runs. We will add it in the revised version. **(3)** (The paper is
rather poorly written) We apologize for the confusion caused
by the writing. We will improve it in the revised version.

Figure 2: The five dashed lines correspond to the multiple runs of GD with the Xavier initialization. It is shown that GD successfully escape the saddle region for only 2 out 5 times in the given number of iterations.

[Meta-Review · NeurIPS 2019]

The reviewers appreciated the work on the initialization even if they deemed it incremental. The experiments on the nonlinear network in the rebuttal was useful and I encourage the authors to expand the experimental section using more realistic setups to show how the theory matters in practice.